# Considerations and Limits of Embedding Sensor Nodes for Structural Health Monitoring into Fiber Metal Laminates

**DOI:** 10.3390/s22124511

**Published:** 2022-06-14

**Authors:** Sarah Bornemann, Walter Lang

**Affiliations:** Institute for Microsensors, Actuators and Systems (IMSAS), University of Bremen, 28359 Bremen, Germany; wlang@imsas.uni-bremen.de

**Keywords:** structural health monitoring, fiber metal laminates, electronics, sensor node, embedded, high temperature

## Abstract

The objective of this article is to present the results of our investigations concerning the environmental conditions that can be expected during the embedding process into fibre metal laminates and the consequences for a sensor node for structural health monitoring. The idea behind this investigation is to determine for which manufacturing conditions the integration of sensor nodes into the material can be done and to identify limits for this. The sensor nodes consist of commercially available integrated circuits and passive components soldered onto an adhesive-less flexible printed circuit board. They are tested under conditions above their specified limits, to find out if they are still working reliably after experiencing 155 min of 180 ${}^{\circ }C$ and 7 bar of pressure. Apart from occurring temperature damage, the effect of surrounding fibres potentially pushing away the components under the amount of pressure of the manufacturing process, as well as the potential of shorts due to conductive fibers are investigated and suitable solutions to prevent this are evaluated. One experiment exceeding the typical requirements of a fiber metal laminate embedding process for structural components will be conducted at 250 ${}^{\circ }C$ for 10 h, in order to determine the limits of embedding electronic sensor nodes. This time and temperature combination is expected to cause irreversible damage to the electronic system. Results show that it is possible to integrate electronics into materials under conditions far above their specifications when precautions are taken but also that there are limits that must not be exceeded during the embedding process.

## 1. Introduction

Fiber Metal Laminates (FML), containing alternating layers of Fiber-Reinforced Plastic (FRP) and metal, offer significant advantages over monolithic aluminum parts. They show better fatigue and impact damage performance [1], which makes them promising materials for aircraft applications. For every component of an airplane, especially load bearing structures, periodic inspection intervals during their lifetime are specified by the airlines. Various maintenance methods exist, reaching from visual inspection, mainly for the detection of surface defects in metallic parts towards radiographic, thermographic, and ultrasonic inspection in miscellaneous implementations to reveal damage inside the material [2]. To assist or even replace existing, extensive methods for the detection of damages inside composite structures and FMLs, there have been several research efforts in the last years. Most elegant approaches to implement a damage detection are directly integrated monitoring systems inside the component, so that no large external aperture is necessary during maintenance. Approaches of integrating sensors or sensor systems into FRP have been investigated in recent years [3,4,5,6,7,8]. Nevertheless, there are only a few such systems integrated inside FMLs. This material comes up with some higher requirements on the sensor nodes to be integrated. Additionally, the material consists of thin material layers which restrict mainly the height of embedded systems. Nevertheless, the challenge whenever embedding sensors or even complete sensor systems into materials is to keep the foreign body effect or wound inside the monitored material as small as possible [9], while at the same time implementing all required functionalities into this limited space. The work of Lynch and Loh [10] provides a good overview of different approaches for sensor nodes and sensor networks and different power supply possibilities for both approaches, attached to the monitored component and material integrated. In the field of embedding miniaturized sensor nodes, Voges et al. [11] presented a monolithic epoxy covered system for smart manufacturing in Industry 4.0.

The main problems during the embedding process of sensor nodes into FML are expected to be temperature, pressure and the surroundings, potentially conductive fibers inside the material itself. Concerning high temperature influences on flash memories, several investigations have been done in the past. Singh et al. [12] investigated a new flash memory cell approach, preventing problems at elevated temperatures like storage charge leakage by design adaptation. Morgul et al. [13] state that flash operations like read and write access are generally less destructive at higher temperatures, but data retention suffers in those environments. Therefore, they recommend the execution of operations on one flash in a warm environment. For data retention, a transfer to a separate flash memory in a cold environment is recommended.

During FML manufacturing, the temperatures depend on the chosen material, which strongly depends on the later purpose of the components. Sensor nodes to be embedded into the material have to withstand this process with intact, previously programmed flash memory content; otherwise, they are not suited to be embedded. To calculate the data retention in floating gate flash memory devices, Equation (Equation 1), documented in [14] and based on the Arrhenius equation, can be used [13,15,16]. The rate of change at temperature $T_{X}$ is expressed as $t_{X}$, $E_{AA}$ is the apparent activation energy, $k_{B}$ is the Boltzmann’s constant, and $A_{F(T_{1},T_{2})}$ is the acceleration factor, which describes the accelerated aging of a flash memory at temperature $T_{2}$ compared to the warranted lifetime stored at $T_{1}$. This will be consulted to estimate a temperature above warranted data retention time of the microcontroller chosen for the sensor node to identify a test scenario under a temperature suspected to limit the functionality of the system:(1)$$A_{F(T_{1},T_{2})}=\frac{t_{1}}{t_{2}}=\exp\left(\frac{E_{aa}}{k_{B}}\;\cdot \;\left(\frac{1}{T_{1}}-\frac{1}{T_{2}}\right)\right)$$

The aim of this study is to identify for which circumstances of FML part manufacturing the approach of integrating digital sensor nodes is feasible and to identify critical conditions for which an external approach is necessary because irreversible damage is expected. In our previous conference contribution [17], a proof of concept was presented, in which a selection of suited components were already put in a similar environment for 190 min. This resulted in no irreversible damage within the investigated active specimen, as well as only slight changes in the values of passive components. However, no statements concerning data integrity of the memories after embedding, complete system functionality nor possible effects of actual FML layers around the sensor node could be made within that study. Therefore, this follow-up investigation is intended to be a final validation of the concept and to provide an orientation for embedding electronic systems with the purpose to serve as Structural Health Monitoring (SHM) sensor nodes into FML.

## 2. Sensor Node for SHM Using Ultrasonic Waves

Requirements for the sensor node are a low power design and flat housings to keep the wound in the material as small as possible. A very fast data acquisition with a high amplification to increase the sensor signal prior to data conversion is required as well. Furthermore, the potential to provide a wireless data and energy link for the sensor node embedded into the component material layers.

The developed sensor node is supposed to be suited to work with two entirely different sensor principles: piezoelectric sensors, as used for several previous investigations on SHM using Guided Ultrasonic Waves (GUW) [18,19,20], as well as with an innovative, piezoresistive Micro-Electro-Mechanical System (MEMS) sensor recently developed for the detection of GUWs in FMLs [21]. Deformation of piezoelectric sensors leads to an accumulation of charges that can be measured as a voltage difference at their electrodes and directly converted by an Analog-to-Digital Converter (ADC) after proper amplification. The piezoresistive sensor on the other hand is designed as part of a Wheatstone-Bridge circuit that needs to be provided with a supply voltage to work. The bridge voltage can then be amplified and digitized by an ADC as well. Therefore, when designing a sensor node working for both sensor approaches, an additional voltage supply for the resistive approach has to be provided, but, apart from adjusting the amplification factor, nothing else has to be changed. Hence, a digital approach was chosen providing an optional power supply and the additional advantage of a frequency independent sensor readout and a potentially wireless transmission of the data using standard protocols.

For the aim of a wireless sensor node using a single antenna, a good trade-off between high data rate and energy harvesting capabilities on one protocol had to be found. With this, the choice was to use the ISO 15693 protocol at 13.56 MHz and using the ST25DV-I2C series which comes with an on board energy harvesting unit, leading to a reduced component count on the sensor node. The ST25DV64K offers 64-kbits of Electrically Erasable Programmable Read-only Memory (EEPROM) that can be filled with measurement data and read out directly afterwards by an external reader device. Energy harvesting capabilities are limited by standard definitions and depending on whether simultaneous communication shall be executed. Deciding that no data has to be transmitted during measurement operation of the sensor node, the Radio Frequency Identification (RFID) energy harvesting from magnetic field can provide up to 7.9 mA at 2 V harvested voltage, resulting in 15.8 mW of theoretically available power for system supply [22].

With this, a supply voltage suited for all components inside the system could be defined. Most low power system architectures on the market work at 3.3 V or even 1.8 V. Given the circumstances of RFID energy harvesting, a system voltage of 1.8 V was chosen, leaving a possible dropout voltage of up to 200 mV for a linear voltage regulator. Additionally to this selection of components, an instrumentation amplifier would be necessary to increase the sensor signal amplitude, as well as a filtering stage to prevent high frequency signals from coupling in. Considering the frequency of the expected GUW signals, a rail-to-rail instrumentation amplifier having a 500 kHz bandwidth, a common mode rejection ratio of 94 dB, and a 2 V/μs^−1^ slew-rate, which additionally is offered packaged in a low-profile housing, was chosen for this. The labeled CAD design (a) as well as a picture of the assembled sensor node (b) under tests in this study can be seen in Figure 1.

The most important specifications of the integrated circuits (IC) and passive components used are listed in Table 1. The components were chosen for this as they are extraordinarily well suited for the application of a wireless, low-power electronic sensor node for SHM using GUWs for the detection of damages.

All components are placed on an adhesiveless, flexible Printed Circuit Board (PCB) made from polyimide and covered with a flexible green coating, with a total thickness of 91 μm, in order to achieve a better structural integrity and a minimized sensor node height. For an even better structural compliance, holes were implemented along the edges of the sensor node to allow resin to flow through and properly surround the sensor node. The functional block structure of the sensor node is shown in Figure 2.

### 2.1. Package Thinning

It can be seen from Table 1 that the maximum height the amplifier is 200 μm above the maximum height of all other components chosen for the sensor node. Therefore, to achieve the smallest height possible, package thinning is done for this part. Different approaches can be considered: mechanically abrasive, laser evaporation, or chemical etching. The mechanical method is easy to use and well suited if only a small part of the package has to be removed. Using a polishing machine, as presented in [23], or simply using sandpaper combined with caliper measurements, to know when to stop, the package can be thinned down before soldering. An optical inspection of the thinned package using a reflected-light microscope will reveal any cut bond wires if the package should have been thinned down too much. With this type of IC, a thinning of up to 100 μm did not result in any damage; experimentally removing more of it resulted in visible bond wires which were apparently separated because the component did not work anymore. The microscopic picture in Figure 3 shows a cut image of the amplifier and reveals the reason for this. The bond wires leading from the silicon die towards the soldering pads can clearly be seen reaching into the upper region of the package. There are about 150 μm of empty package space that can be removed without losing functionality. As the chosen method is not material selective, stopping already at 100 μm is a good choice. Reducing component height by this amount is a good start, improvements of package thinning will be investigated further, probably using the chemical approach.

### 2.2. System Performance of the SHM Sensor Node

The general performance of the developed sensor node has to be tested before thinking of the embedding process itself. Therefore, a measurement of the current consumption during the sensor signal processing task was performed using the Keithley DMM6500 digital multimeter. This resulted in an average current of 5 mA over the whole active time and 6.1 mA during ADC sampling. For the system powered with 1.8 V, this results in an average power consumption of 9 mW and 11 mW during measurement.

Additionally, the measurement performance regarding amplification and sample rate for a generated sensor signal was tested. Therefore, a ST25R3916-DISCO reader development board was used to read out the complete EEPROM of the sensor node after measurement. Data received in ASCII-format could then be plotted using Matlab. A schematic representation of the measurement setup is shown in Figure 4. One result can be seen in Figure 5. It shows a measurement of a 100 kHz signal of 3 mV after amplification with G = 213 and digitization. The top part shows the measurement over the whole duration, the bottom part is zoomed to the first 100 sample values. The input signal is generated by the PICOSCOPE 2206B function generator. In the zoomed part, 7.5 cycles can be seen which are equal to 75 μs. The sampling rate can thereby be calculated to 1.33 Msps. With this result, sample rate and amplification are high enough to sample the expected sensor signals used for SHM. With this, the SHM sensor node is ready to be tested under conditions that occur during the embedding process into FML.

## 3. Considerations When Embedding Sensor Nodes into FML

### 3.1. Process Parameters for FML Manufacturing

The manufacturing process parameters for FML components mainly depend on the recommended curing cycles of the used prepreg material. In Table 2A–G, a small selection of different curing recommendations is listed, focusing on materials for aerospace applications as recommended by prepreg manufacturer HEXCEL^®^. The listed prepreg resin systems are available with woven, multi-axial or unidirectional reinforcing fibers. It can be seen that materials are available in a wide range of curing conditions. In general, materials with high temperature curing processes are better suited for applications with higher demands after curing. Concerning the matrix type, epoxy-based matrix systems clearly dominate the market due to their good mechanical performance. The product that will be used for embedding tests will be HexPly^®^ 8552, reinforced with unidirectional AS4 carbon fibers (as 12 K tow) with a filament diameter of 7.1 μm because it is the “preferred product for aerospace structures” [24] stated by the manufacturer. Therefore, the first experiment “Miniature autoclave system test” will be conducted, simulating real embedding conditions for this material around the sensor nodes without actually using prepreg material around them, to be able to visually inspect and measure the sensor nodes afterwards.

As there are is a small number of materials cured at more challenging conditions, e.g., Bismaleimide resin (see Table 2G), the material itself is recommended to be cured at 191 ${}^{\circ }C$ but requires a 16-h long, 232 ${}^{\circ }C$ post curing (PC) cycle. The possibility to embed sensor nodes in these materials will be investigated in a second “Bismalemide embedding conditions system test” experiment in Section 3.4 as well.

### 3.2. Miniature–Autoclave System Test

To execute the miniature–autoclave test, the same setup as already presented in [17] is used. For this experiment, complete working sensor nodes are tested.

Four sensor nodes are prepared, presented in Figure 6a and put into service. Measurements of sample signals at different frequencies are measured before the test to compare the results before and after it. The test samples are then put on an aluminum plate and put into the miniature–autoclave. The plate is slightly elevated on a U-shaped sealing tape. This enables the setup to be put in vacuum from the connection beneath the plate and allows a simultaneous temperature measurement directly under the plate. The measurement is done using a Voltcraft TP-202 K-type thermocouple, connected to a Keithley DMM6500 digital multimeter. The connection leading towards the thermocouple is partly led through the vacuum tubing to place it as close to the sensor nodes as possible. Medium vacuum of 0.15 bar is applied through the bottom connection of the steel box while an additional 7 bar of pressure is applied to the samples through the side connection.

The temperature measured during the miniature–autoclave system test can be seen in Figure 7 presented by the red, solid line. It shows that the experiment lasted for 340 min while the test specimens were exposed to the recommended curing temperature for most available resin systems of at least 180 ${}^{\circ }C$ for 155 min, reaching top temperatures of up to 186 ${}^{\circ }C$ temporarily. This temperature over the tested duration covers manufacturing stresses occurring due to temperature and pressure influence for most available prepregs. Curing process temperatures of materials A–F from Table 2 are presented by the dashed lines. Therefore, this experiment is appropriate to verify the survival of the sensor node after an embedding process into epoxy and phenolic resin based prepregs.

### 3.3. Results after Miniature–Autoclave System Test

The results of the miniature–autoclave experiment show that all four sensor nodes remain completely functional and undamaged after this test. Only a slight color difference of the flex PCB covering layer can be observed. Apart from this change, the influence of temperature and pressure seems to be negligible.

As already stated before, ideally, the manufacturing process should not decrease the expected lifetime of the sensor node so much that it is less than the expected lifetime of a structure whose health it should monitor. Therefore, the lifetime reduction of flash memory after the recommended embedding process into HexPly^®^ 8552 [30] is calculated using Equation (Equation 1) and $E_{aa}$ = 1.1 eV as recommended in [16]. Assuming medium heat up (2 ${}^{\circ }C$ per minute) and cool down (3.5 ${}^{\circ }C$ per minute) rates, the whole manufacturing process starting at 24 ${}^{\circ }C$ and ending at 60 ${}^{\circ }C$ lasts for 295 min. Dividing the curing process into two steps for ease of calculations, $S_{1}\leq$ 105 min with T = 110 ${}^{\circ }C$ and $105<S_{2}\leq$ 295 min with T = 180 ${}^{\circ }C$, this results in $A_{F(85,110)}$ = 10.23 for the first segment and $A_{F(85,180)}$ = 1758 for the second segment. This results in a combined lifetime reduction for the flash memory of 5591 h. From the data sheet of the microcontroller, the data retention for program memory is warranted for 30 years for a storage temperature of 85 ${}^{\circ }C$ [32]. At an operating temperature during lifetime below 85 ${}^{\circ }C$, the specified data retention time still remaining is thereby 257,209 h or 29.36 years.

From this experiment, we expect that the designed sensor nodes can be safely embedded into the target material with only a negligible lifetime reduction and without any failures due to curing conditions. Being designed as a health monitoring system for structural components, this result is satisfying so far. However, considering future applications, the identification of embedding conditions that is too challenging for the designed sensor node is the goal of the following test.

### 3.4. Bismalemide Process Temperature System Test

In contrast to the first experiment that was mainly done for epoxy-based resins, there are some types of prepreg resins that are supposed to be cured or post cured at higher temperatures. Table 2 lists one bismaleimide based prepreg (G) as an example. A calculation of lifetime reduction for an embedding of sensor nodes into this material using Equation (Equation 1) would result in an exceedance of the data retention time by theoretically 28 years, taking only the post curing process into account.

To see if this really results in a system failure of the sensor node, a high temperature experiment will be conducted using only temperature as the stress source. Transposing Equation (Equation 1) to $T_{2}$ results in Equation (Equation 2). The temperature needed to emulate the end of data retention warranty where bit errors might be observable when reading out the flash memory content can be calculated from this:(2)$$T_{2}=\frac{1}{-\frac{k_{B}}{E_{aa}}\;\cdot \;\ln\left(A_{F(T_{1},T_{2})}\right)+\frac{1}{T_{1}}}$$

Using the equation provided above and assuming $E_{aa}$ = 1.1 eV as recommended in [16] and an accelerated aging to end of data retention specified after 10 h, resulting in $A_{F(T_{1},T_{2})}$ = 26,280, the resulting temperature equals 501.28 K or 228.13 ${}^{\circ }C$. Therefore, we assume that, after an even higher stress of 10 h inside the oven at 250 ${}^{\circ }C$, a functional failure can be expected and memory differences may occur in flash memory. Given the fact that the lifetime of an SHM sensor node should be ideally longer than the lifetime of the part the SHM system is designed for, the temperature and time over which the ICs are exposed to these conditions should be even lower. To investigate if there are memory differences detectable after the experiment, two sensor nodes are prepared and put into service. The same measurements as before are done to assure the functionality before testing, and flash memory content is read out for later comparison. The sensor nodes are put into a *Heratherm OMH60-S* oven at 250 ${}^{\circ }C$ for 10 h. The results are described in the following section.

### 3.5. Results of the Bismalemide System Test

A first intermediate measurement after 4.5 h was done to check whether a shorter exposure to this temperature results in less effect on the system. Following Equation (Equation 1), the flash memory content should still be intact after this time. After cool down, the sensor nodes were operated via the wireless link in the same procedure as previous to the high temperature test. Results show that, after 4.5 h, the controlled voltage, signal amplification as well as the RFID EEPROM readout worked well, but the measurement values were not updated anymore. The I2C bus connection did not work but the GPIO pin was toggled as it was supposed to. Both sensor nodes were again connected to the debugger, memory was read out and compared to the memory content before the test. No changes were detectable inside flash memory, and the content was completely unaltered after 250 ${}^{\circ }C$ for 4.5 h. The debugging environment still attached, one microcontroller was flashed again and a step by step run-through was done showing that even ADC conversion and writing values directly to the internal memory was still possible. The fault in code occurred while trying to establish the I2C connection between microcontroller and RFID EEPROM. As not even a clock-signal was generated by the microcontroller for I2C communication start, the cause of this error is presumably somewhere within the hardware abstraction layer, but this was not investigated any further.

The sensor nodes were both put back into the oven for another 5.5 h and measurements were attempted the same way as before. Sensor node 1 was not able to be found by the reader anymore, so no readout of measurement values was possible. Additionally, no voltage harvested from the reader field was measurable. Therefore, it is assumed that this long temperature stress caused damages to the RFID IC. Applying an external voltage to the voltage regulator revealed that the functionality of this part was still completely available and a stable voltage of 1.855 V was generated at the output. Sensor node 2, on the other hand, was still able to be found by the reader and was able to generate 1.857 V when an external voltage was provided as well as the same amount harvested from the reader’s field. However, comparable to the intermediate test after 4.5 h, the measurement values were not properly updated. Both microcontroller flash memories were read out again with the result that no difference could be detected in memory content before and after the high temperature test once more. Therefore, the same assumption as for the result after 4.5 h at 250 ${}^{\circ }C$ has to be made. Apart from the decreased functionality after the second experiment, the material used for the flexible PCBs is clearly not designed to withstand such temperatures for a longer duration. A photo of one sensor node after temperature test can be seen in Figure 8. After the test specimen was taken out of the oven, the flexible top coat was quite brittle and could be easily scraped off the conductor tracks.

Nevertheless, this experiment clearly shows that an embedding process at such high temperatures inevitably causes harm to the function of the designed sensor node. For components requiring curing cycles containing comparable conditions, an embedding during part manufacturing cannot be recommended.

### 3.6. Electrical and Mechanical Considerations When Embedding Sensor Nodes into FML

Apart from the previously tested influences of pressure and temperature during the embedding process into FML during component-manufacturing, there are further considerations to be made. One challenge is the conductivity of surrounding materials. There are of cause the metal layers, but, depending on the fiber type, in case of carbon fibers, even the FRP layers can be electrically conductive. This may lead to electrical short circuits on the sensor node disabling functionality. There are several ways to prevent this:Insulating varnish spray;3D printed housing;UV glue;Polyimide tape;Encapsulating with epoxy resin.

A 3D printed housing would be one easy approach to protect the sensor node from surrounding fibres and available materials meanwhile offering even high temperature compatible choices. The disadvantage is that there will be a large increase of thickness by simply putting a housing around the complete sensor node. Using insulating varnish is an easy to use process which results in a homogeneously distributed, thin layer of insulation, but this offers no mechanical stabilization for the soldering connections and there is only a small choice of highly temperature resistant insulating varnishes. Advantages with respect to heat resistance can be expected using polyimide tape but again no sufficient mechanical stabilization is created by this. In an experimental embedding without mechanical stabilization of the components, only half of the embedded specimens were functional after embedding. Therefore, we assume that, at given temperature and pressure influences during the embedding process, some kind of mechanical stabilization should be provided around the components. The best results were obtained with a thin encapsulation of the sensor nodes with *LOCTITE^®^ EA 9596 AERO* epoxy film adhesive. Preferably, the same type of resin already present within the prepreg material should be used for the encapsulation in future experiments to ensure optimal material compatibility.

## 4. Discussion

Results presented in this paper present considerations and limits for the embedding of sensor nodes into FML components manufactured in prepreg-based autoclave processes. Slightly thinning bigger components and encapsulating the sensor node into an epoxy cover layer should be done before embedding, to achieve the best performance. Practical experiments resulted in a comparably high failure rate among sensor nodes not properly protected electrically as well as mechanically. One failed FML plate containing glass fibers in which an unprotected sensor node was embedded was de-metallized using wet chemical etching with hydrochloric acid. The result revealed the sensor node inside the glass fiber layers and, using transmitted light through the plate thickness, the positions of components could be roughly estimated. A comparison to the original component positions was made by superimposing a highly transparent image of the CAD design, but no significant shift of components could be observed. Nevertheless, the electrical contacts of the components could not be tested, and only half of the unprotected nodes were working after embedding, while all four material-integrated test specimens previously properly embedded into a protective epoxy layer survived the process unharmed. Therefore, a protection against electrical and mechanical influences is highly recommended.

Considering temperature influence, depending on the type of resin, FML manufacturing results in vastly varying curing conditions. Sensor nodes to be embedded into FML components during manufacturing inevitably have to deal with these requirements. Experimental investigations show that common curing cycles for epoxy-based resins, which are the most commonly recommended prepreg matrix for structural aircraft components, result in no reduction of the sensor node functionality. Even the warranted lifetime of the flash memory inside the microcontroller, which is assumed to be the most heat sensitive component in the design, is barely even reduced by the recommended curing cycle of HexPly^®^ 8552, as calculations demonstrate.

In contrary to epoxy-based resins, special high temperature resistant prepregs based on e.g., bismaleimide resin are not suited for sensor node integration—at least not using standard commercially available components with integrated flash memory. Apart from the influence of elevated temperatures during post curing on the flexible PCB, calculations show that the required long time, high temperature post curing processes lead to increased reaction rates inside the architecture of flash memories. Taking the warranted lifetime into account and using Equation (Equation 1), the end of life is reached or even already exceeded after the post curing procedure is done. Therefore, it can be concluded that, for sensor nodes that should be embedded into high temperature resistant FML components, further improvements to the heat resistance of commercially available electronic circuits would have to be done. However, sensor node development for an integration into epoxy based FML for SHM purposes using standard components with careful encapsulation before embedding is a feasible approach for the improvement of today’s maintenance work at airports. Thus, even though embedding temperatures exceed the given component specifications, destruction is not expected and was not observed within the investigated limits.

Following the presented validation of suitability to be embedded into FML, sensor nodes with attached sensors were already successfully embedded into FML following the presented procedure. In future investigations, GUW inside the material will be measured and measurement data will be wirelessly transmitted for SHM evaluation purposes. For measurement evaluation, reversible, artificial damage will be introduced to the investigated components and compared to the signal without any damage.

## 5. Conclusions

We presented our investigations regarding the feasibility of embedding sensor nodes made from commercially available components into FML. Results show that an integration is possible even under conditions far above the IC specifications, e.g., temperatures of up to 180 ${}^{\circ }C$ and pressures of up to 7 bar, if certain precautions are taken. However, it was also experimentally validated that there are limits to this that must not be exceeded to successfully embed sensor nodes into FML.

## Figures and Tables

**Figure 1 sensors-22-04511-f001:**
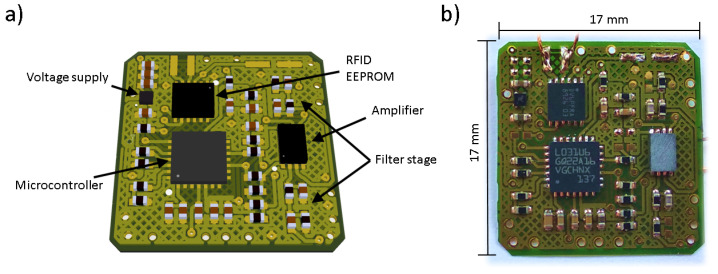
Sensor node for embedded GUW sensor connection. (**a**) sensor node from CAD with main part description; (**b**) photograph: manufactured and assembled PCB.

**Figure 2 sensors-22-04511-f002:**
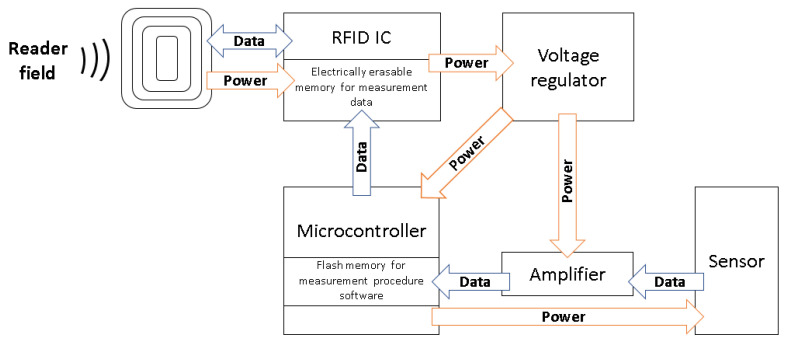
Functional block overview of the sensor node.

**Figure 3 sensors-22-04511-f003:**
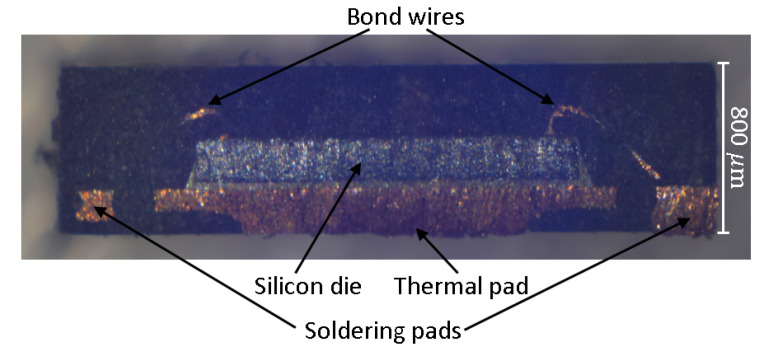
Cut image through the amplifier. The silicon die with one clearly visible bond wire (right side) and another bond wire barely visible (left side).

**Figure 4 sensors-22-04511-f004:**
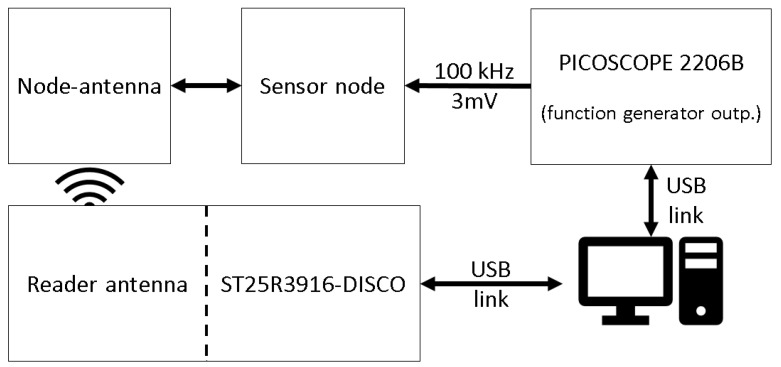
Schematic measurement setup of the system performance tests.

**Figure 5 sensors-22-04511-f005:**
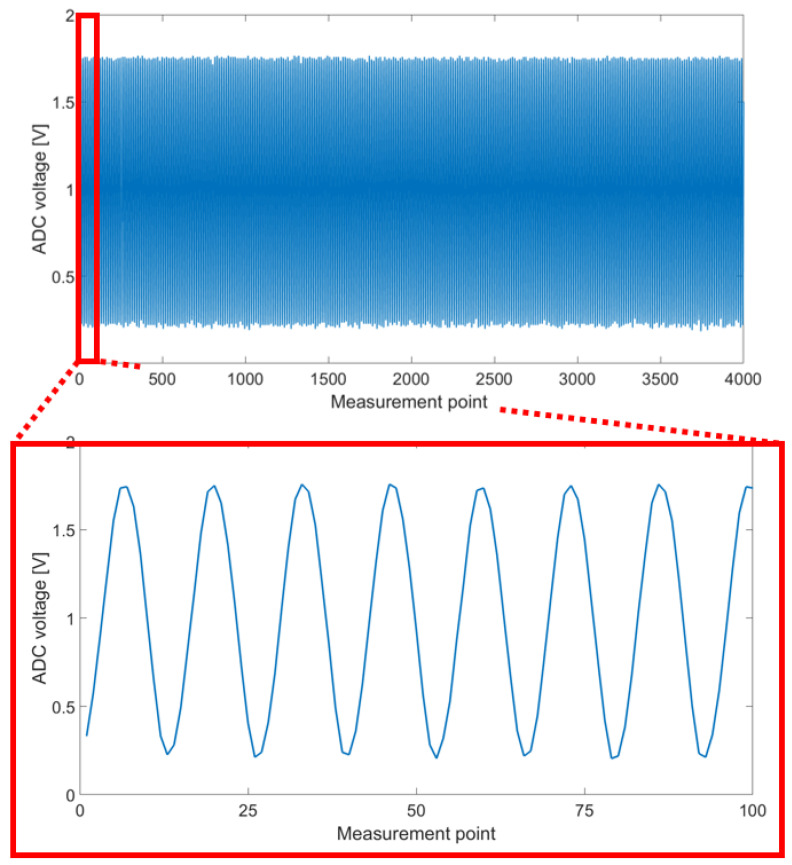
Measurement of a 100 kHz, 3 mV sine wave amplified and digitized by the sensor node, data plotted in Matlab.

**Figure 6 sensors-22-04511-f006:**
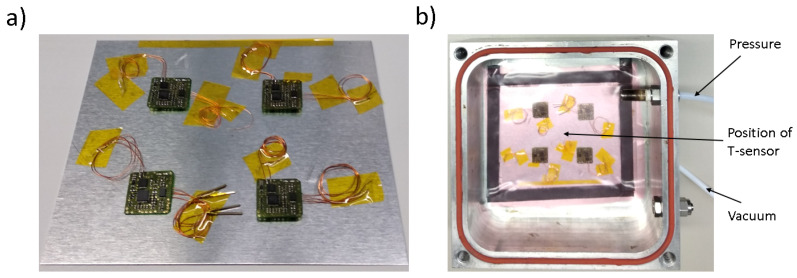
Experimental setup for sensor node temperature tests. (**a**) sensor nodes on aluminum plate; (**b**) sensor nodes inside a miniature autoclave with vacuum bagging, vacuum and pressure.

**Figure 7 sensors-22-04511-f007:**
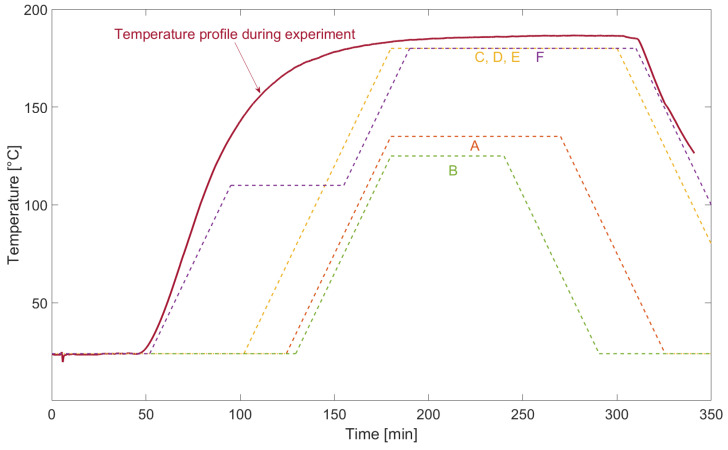
Temperature profile inside miniature autoclave for system temperature conformity tests. A–F recommended curing cycles for prepregs (see Table 2).

**Figure 8 sensors-22-04511-f008:**
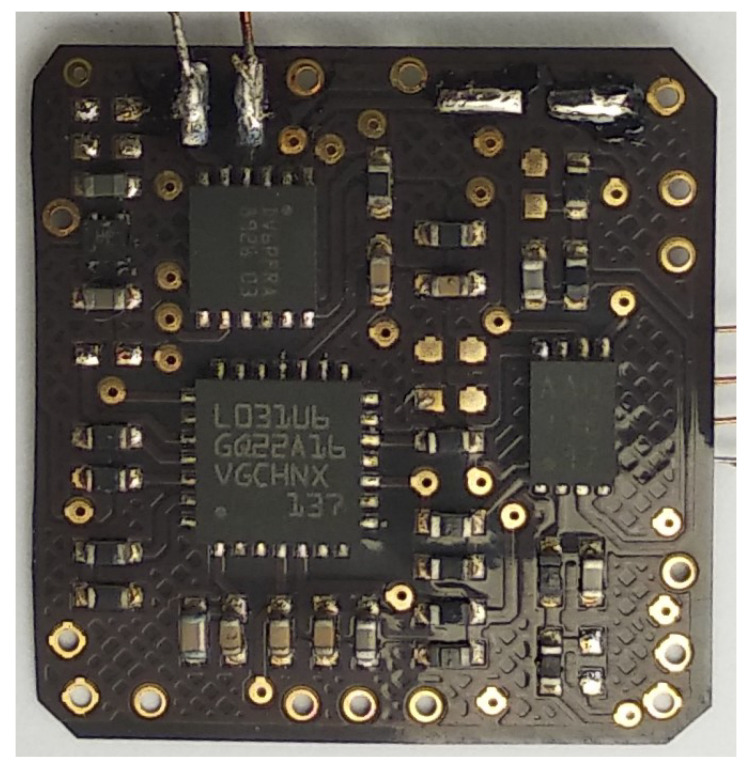
Photograph of one sensor node after a high temperature stability test.

**Table 1 sensors-22-04511-t001:** Components for a wireless SHM sensor node for GUW based damage detection.

Component Purpose	Component Number	Key Features	Max. Package Dimensions
Voltage supply	TPS7A03185PDQNR	1.8V, 200 mA 0.27V max. dropout −65–150 ${}^{\circ }C$ storage	1.05 × 1.05 × 0.4 mm^3^
Microcontroller	STM32L031G6U6	1.65V min. voltage Low-power architecture 1.14 Msps, 12 bit ADC −65–150 ${}^{\circ }C$ storage	4.1 × 4.1 × 0.6 mm^3^
RFID EEPROM	ST25DV64K-JFR6D3	ISO/IEC 15693 1 MHz I2C −65–150 ${}^{\circ }C$ storage	3.1 × 3.1 × 0.6 mm^3^
Amplifier	MCP6N11	CMRR = 115 dB Bandwidth = 500 kHz Slew rate = 2 V/μs^−1^ Supply-I, typ = 800 μA −65–150 ${}^{\circ }C$ storage	2 × 3 × 0.8 mm^3^
Resistances	several types	0402 packaging −55–125 ${}^{\circ }C$ operating	1.1 × 0.55 × 0.4 mm^3^
Capacitances	several types	0402 packaging −55–125 ${}^{\circ }C$ operating	1.1 × 0.55 × 0.55 mm^3^

**Table 2 sensors-22-04511-t002:** Curing conditions for a selection of HEXCEL HexPly^®^ prepregs recommended for aerospace application (component thickness < 15 mm).

No.	Prepreg	Temperature	Pressure	Duration	Material	Domain
A	200 [25]	150 ${}^{\circ }C$	3–7 bar	30 min	Phenolic	Interiors
B	913 [26]	125 ${}^{\circ }C$	7 bar	60 min	Epoxy	Fairings
C	922-1 [27]	180 ${}^{\circ }C$	3–5 bar	120 min	Epoxy	Engine and nacelle
D	M18 [28]	180 ${}^{\circ }C$	7 bar	120 min	Epoxy	Space applications
E	M21 [29]	180 ${}^{\circ }C$	7 bar	120 min	Epoxy	Primary structures
F	8552 [30]	180 ${}^{\circ }C$	7 bar	120 min	Epoxy	Structural parts
G	F655 [31]	191 ${}^{\circ }C$ + PC 232 ${}^{\circ }C$	5.86 bar	240 min + PC 16 h	Bismaleimide	Engine parts

## Data Availability

Data presented in this study are available on request from the corresponding author.

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
