# Peer review of "Considerations and Limits of Embedding Sensor Nodes for Structural Health Monitoring into Fiber Metal Laminates"

_sensors, 2022, doi:10.3390/s22124511_

Round 1

Reviewer 1 Report

1. Use “Microscopic” picture instead of the “Microscope” picture.

2. Please provide a schematic diagram to explain the system performance of the SHM sensor mode. This may provide more clarity to the process. Authors can also think of adding image(s) of the setup.

3. Provide reference(s) for the formulas used in the paper.

4. Authors should provide suitable references for the already known facts used for explanations in the results section. 

5. The language of the manuscript is good, however, there are few errors which need to be corrected.

Author Response

First of all, thank you very much for your time and  comments on the paper!

  1. Use “Microscopic” picture instead of the “Microscope” picture.

Reply to 1: Changed it in the text.

  1. Please provide a schematic diagram to explain the system performance of the SHM sensor mode. This may provide more clarity to the process. Authors can also think of adding image(s) of the setup.

Reply to 2: Added a schematic diagram (Figure 4). Also took photos of the setup but they don’t show that much, mainly the development board and the PicoScope with the much smaller sensor node next to them, the antenna of the sensor node is buried beneath the dev.-board one. For comprehensive purposes I think the schematic representation helps much more at this point.

  1. Provide reference(s) for the formulas used in the paper.

Reply to 3: References on the recommendation to use this formula as well as the source of the formula itself were added to the text and the text around citations is highlighted (\hl{}).

  1. Authors should provide suitable references for the already known facts used for explanations in the results section. 

Reply to 4: Reference recommending to use Eaa=1.1 eV for NAND flash memories has been added and highlighted in line 141.

  1. The language of the manuscript is good, however, there are few errors which need to be corrected.

Reply to 5:  Critically read through the document again and corrected several mistakes and highlighted the changes using \hl{ }. I hope I have found all incorrect parts.

Reviewer 2 Report

 This work has merit and addresses an important topic. A few comments/questions are as follows:

Comment 1:

lines 9-11:The authors state in the abstract,

"Apart from occurring temperature damage, the effect of surrounding fibres potentially pushing away the components under the amount of pressure of the manufacturing process, as well as the potential 1 of shorts due to conductive fibers are investigated and suitable solutions to prevent this are evaluated." 

It is recommended the authors add additional comments in the Discussion (Section 4) on the  effect of fibers potentially pushing away the components during the manufacturing process.  Is there any evidence that may fibers push around component may have occurred?

Comment 2:  

Figure 2:  Define or add a description of "HF" and "EEPROM" on figure or in figure description caption.

Comment 3:

Figure 4:  The inset dotted line is obstructing a portion of the text including x-axis (1000) and "Measurement point". It is recommended to revise figure. 

Comment 4:

lines 128 to 140: For the prepregs used in Table 2, what are the fiber orientations? This is recommended to be included in either table or the text.  Also, although used for aerospace, why were these fiber orientations chosen?  What are fiber diameters?   It is recommended to comment on these aspects.

Comment 5:

Section 4. Discussion,. It is recommended to split concluding statements into a "Conclusions" section

Author Response

First of all, thank you very much for your time and  comments on the paper!

Comment 1: lines 9-11:The authors state in the abstract,

"Apart from occurring temperature damage, the effect of surrounding fibres potentially pushing away the components under the amount of pressure of the manufacturing process, as well as the potential 1 of shorts due to conductive fibers are investigated and suitable solutions to prevent this are evaluated." 

It is recommended the authors add additional comments in the Discussion (Section 4) on the effect of fibers potentially pushing away the components during the manufacturing process.  Is there any evidence that may fibers push around component may have occurred?

Reply to 1: That is a very good point, that explanation got lost in the process. A paragraph (marked with \hl{}) was added to Section 4 with an explanation of the attempt to investigate the reason behind the malfunction of unprotected nodes. Unfortunately, this investigation was done with a previous version of the sensor node with a different design (same components though) and the image quality is not that good due to the surrounding fibers so this image was not added to the text, but a longer explanation instead.

Comment 2:  Figure 2:  Define or add a description of "HF" and "EEPROM" on figure or in figure description caption.

Reply to 2: Changed HF to Reader field and EEPROM to Electrically erasable memory to increase figure comprehensibility even without description, but at the same time keep the memory type name in so it is clear that there are two different memory architectures inside these components.

Comment 3: Figure 4:  The inset dotted line is obstructing a portion of the text including x-axis (1000) and "Measurement point". It is recommended to revise figure. 

Reply to 3: The figure has been revised so that the line is behind the text objects now.

Comment 4: lines 128 to 140: For the prepregs used in Table 2, what are the fiber orientations? This is recommended to be included in either table or the text.  Also, although used for aerospace, why were these fiber orientations chosen?  What are fiber diameters?   It is recommended to comment on these aspects.

Reply to 4: The listed prepregs in Table 2 are available with woven, multi-axial or unidirectional fiber orientations as well as a selection of fiber types from HEXCEL. The chosen material we used for our embedding experiments is HEXPLY 8552 unidirectionally reinforced with AS4 carbon fibers (as 12K tow) with a filament diameter of 7.1 microns. I added the information separately, explaining the availability of different fiber direction choices in lines 132/133 and explaining our chosen prepreg reinforcement further down the text in line 138/139.

Comment 5: Section 4. Discussion,. It is recommended to split concluding statements into a "Conclusions" section

Reply to 5: Added a short Conclusions section with the most important statements in condensed form at the end.
